# Protocol for a realist evaluation of Recovery College dementia courses: understanding coproduction through ethnography

Linda Birt [1,2] Juniper West [3] Fiona Poland,[1] Geoff Wong,[4] Melanie Handley,[5] Rachael Litherland,[6] Corinna Hackmann [3] Esme Moniz-Cook [7] Emma Wolverson,[8,9] Bonnie Teague,[10] Ruth Mills,[11] Kathryn Sams,[11] Claire Duddy [12] Chris Fox[13,14]

For numbered affiliations see end of article.

**Correspondence to**
Dr Linda Birt;
linda.birt@uea.ac.uk

## ABSTRACT

**Introduction** Support following a dementia diagnosis in the UK is variable. Attending a Recovery College course with and for people with dementia, their supporters and healthcare professionals (staff), may enable people to explore and enact ways to live well with dementia. Recovery Colleges are established within mental health services worldwide, offering peer-supported short courses coproduced in partnership between staff and people with lived experience of mental illness. The concept of recovery is challenging in dementia narratives, with little evidence of how the Recovery College model could work as a method of postdiagnostic dementia support.

**Methods and analysis** Using a realist evaluation approach, this research will examine and define what works, for whom, in what circumstances and why, in Recovery College dementia courses. The ethnographic study will recruit five case studies from National Health Service Mental Health Trusts across England. Sampling will seek diversity in new or long-standing courses, delivery methods and demographics of population served. Participant observations will examine course coproduction. Interviews will be undertaken with people with dementia, family and friend supporters and staff involved in coproducing and commissioning the courses, as well as people attending. Documentary materials will be reviewed. Analysis will use a realist logic of analysis to develop a programme theory containing causal explanations for outcomes, in the form of context-mechanism-outcome-configurations, at play in each case.

**Ethics and dissemination** The study received approval from Coventry & Warwickshire Research Ethics Committee (22/WM/0215). Ethical concerns include not privileging any voice, consent for embedded observational fieldwork with people who may experience fluctuating mental capacity and balancing researcher 'embedded participant' roles in publicly accessible learning events. Drawing on the realist programme theory, two stakeholder groups, one people living with dementia and one staff will work with researchers to coproduce resources to support coproducing Recovery College dementia courses aligned with postdiagnostic services.

## STRENGTHS AND LIMITATIONS OF THIS STUDY

⇒ This study is the first to examine postdiagnosis Recovery College dementia courses run within National Health Service mental health Trusts.
⇒ The data collection methods have been discussed and agreed with a stakeholder group of people living with dementia.
⇒ Using a realist approach has the potential to provide understanding on contextual factors which affect outcomes.
⇒ To avoid privileging subgroups the variety of people involved in coproducing and codelivering Recovery College dementia courses such as staff, people with dementia and family supporters, will be analysed separately and then combined to provide a full picture.
⇒ Access to case sites may be limited due to post-COVID-19 impacts on current provision of Recovery College dementia courses.

## INTRODUCTION

Receiving and adjusting to a diagnosis of dementia is often life-changing for the person and their family, bringing many uncertainties, compounded by social stigma.[1]

The progressive nature of dementia in limiting communication and cognition brings anxieties for people trying to preserve identity and confidence in roles, relationships and interactions, and the person and their family may experience shame associated with self-stigma.[2 3] The postdiagnosis period is critical to adjustment[4]; however, postdiagnosis support globally varies enormously.[5]

In the UK, most people with dementia are assessed and diagnosed within specialist National Health Service (NHS) Mental Health Trust provided memory services following a referral from their general practitioner. Following a diagnosis of dementia,

people should be offered support tailored to meet their individual needs.[5–7] Some people do receive such support; however, there is significant variability in the quality, duration and eligibility of postdiagnostic support across the UK, and what support there is, generally remains poorly defined or evidenced.[7 8]

One form of postdiagnostic support is through Recovery College courses. Recovery Colleges, delivered through NHS Mental Health Trusts and community organisations, support people holistically through individual mental health recovery by focusing on peer-led education and training. This is designed to complement existing clinical interventions and care.[9] The ethos of the Recovery College model is not to frame the word 'recovery' as a clinical 'cure' for mental illness[10] but rather to improve outcomes for people beyond a purely medical model focus on symptom reduction, to help people rebuild meaningful, satisfying lives, despite limitations caused by mental health difficulties.[10 11] Courses may be delivered either as a standalone session or across several sessions. All courses are codesigned and codelivered—coproduced—by people with lived experience and healthcare professionals (staff). People are signposted to self-select and enrol, rather than being referred. In general, any service user, their supporters or staff member can enrol on any course although this can differ between Recovery Colleges. All attendees are considered 'students', as Recovery Colleges aim to use language to empower their attendees and place all at an equal level.

Increasingly Recovery Colleges are coproducing courses specifically about living well with dementia. Recovery Colleges' courses and content may vary; however, they systematically reference and apply a robust conceptual framework which includes five domains of personal recovery—CHIME: Connectedness, Hope, Identity, Meaning and Empowerment.[12] See online supplemental file 1 for further details of the Recovery College model. Such courses align with the strategic health service national dementia strategy aim to empower people living with dementia and their supporters to not only manage their symptoms but also to live positively or 'well' with dementia.[13] Nonetheless, even when there is broad acceptance of a recovery 'approach' to dementia to support finding ways of living well alongside the diagnosis,[14 15] the use of the word 'recovery' used together with dementia does not appear to be acceptable to all staff working in memory services.[15] We acknowledge concerns about the term 'recovery' in the context of dementia, and anticipate this study will contributing evidence towards the continuing wider debate[16 17] through exploring the perceptions of people with dementia and family supporters. The term requires further exploration critically foregrounding the perspectives of people with lived experience among whom the mental health personal recovery approach originated.[18]

The provision of courses for people living with dementia within the Recovery College model is variable across the UK.[19] The impact of the COVID-19 pandemic on delivery of Recovery College dementia courses also appears significant with some courses ceasing and others moving to online delivery, although latterly there is a move back to face-to-face or hybrid delivery methods.[15] Little is known about how the recovery model within the Recovery College works for people with dementia. Evaluation is needed to determine what works for who, how and when, and therefore, this ethnographic study is grounded within a wider project which draws on realist evaluation approaches.

## Aims and objectives

This research project will use participant observation, interviews and review of documentary evidence across five case sites to examine what happens between people living with dementia, their family and friend supporters and staff. Interpreting data through the lens of a realist approach will enable us to address questions about why and how and in which way dementia courses are coproduced and the outcomes for peer tutors and those attending the courses.

## Research question

'What factors support the coproduction and successful delivery of Recovery College dementia courses: what works, for whom, in what contexts and why?'

### Objectives

1. To examine if the characteristics of coproducers lead to difference in outcomes for attendees.
2. To examine if the content and delivery of course material leads to differences in outcomes for attendees.
3. To examine if the recovery college has different outcomes for different types of attendees.

## METHODS AND ANALYSIS
### Study design

We will conduct a realist evaluation[20] using a case study design to develop a realist programme theory, to understand how mental health service-delivered Recovery College courses currently lead to intended and unintended outcomes for people living with dementia, their family and friend supporters and staff. This work is the second step of a research project called DiSCOVERY (reference NIHR131676) with planned start and ends dates 1 January 2022 to 31 December 2024. Learning from this research will support understandings on the implementation of a complex health service provision in different social and service contexts. It is likely to provide knowledge on how to optimise future course development and implementation on a national level.

The initial programme theory was developed from a realist review (manuscript under preparation). A case study approach is appropriate for understanding tightly bounded contexts, by place, time and actors, within naturally occurring events.[21] Here, a case is a Recovery College course offered to people following a diagnosis of dementia in English NHS Mental Health Trusts. The

case is bounded by the time from beginning of coproduction through delivery to evaluation of a single course. The place is either through online meeting or in person and the actors are any NHS staff involved in supporting Recovery Colleges; staff, people with dementia and family or friend supporters involved in the commissioning, coproduction, delivery or evaluation of the course and any person with dementia, supporters or staff who attend the course. Data collection is ethnographic observation and questions, realist interviews and review of documents relating to the courses.

## Patient and public involvement: coproduced research

In accordance with the ethos of coproduction inherent in Recovery Colleges, in this study, we are working throughout the whole programme of research with a diverse patient and public involvement (PPI)—the Partners in Research—advisory group consisting of nine individuals; seven people living with dementia and two people who identify as family carers/supporters. Within the group, two people have been involved in coproducing and delivering a recovery college dementia course. The group's activity is supported by Innovations in Dementia (http://www.innovationsindementia.org.uk/) and the membership has the potential to be flexible as people's wish to be involved may change over the 36 months of the study. The advisory group have been instrumental from the very beginning in developing the research funding application through to refining the aim and considering the practical and ethical challenges of undertaking ethnography in this context. Through a regular series of online meetings, we are coproducing and refining our initial programme theory with our two key stakeholder advisory groups: PPI and staff from memory services and/or involved in Recovery College dementia courses. The PPI advisors in particular are involved in coproducing accessible materials for results dissemination throughout each study work package.

## Recruitment

We plan to recruit up to five case sites across England sampling for the following case characteristics: are running a dementia course in person or online, have experience of delivering Recovery College courses in dementia or are delivering a first course. Case sites will additionally be approached to discuss aspects of cultural inclusion within their recovery course thinking, as reflected within the specific needs and diversity of their local populations. Heterogeneity in cases is important to understand a range of experiences to inform later work to coproduce relevant guidance documents. 'A case' needs to meet two requirements:

i. NHS Mental Health Trust provided Recovery College course on living well with dementia postdiagnosis in England.
ii. Availability of all stakeholder groups involved in the Recovery College dementia course, namely staff, people with dementia, family/friend supporters involved in (a) coproduction and/or delivery and (b) as course attendees.

## Sample

A purposive sample will include characteristics highlighted through our realist review as most relevant for developing the programme theory. We will sample across all stakeholders see figure 1, and across methods of delivery online and in person. Reflecting the diversity of the populations Recovery Colleges serve and the types of courses they provide, will enable analyses to reflect and illuminate developments in different contexts. Based on our realist review these are:

### Inclusion criteria

► NHS Recovery Colleges which support people with dementia accessing memory services, including people from diverse ethnic and cultural local populations,

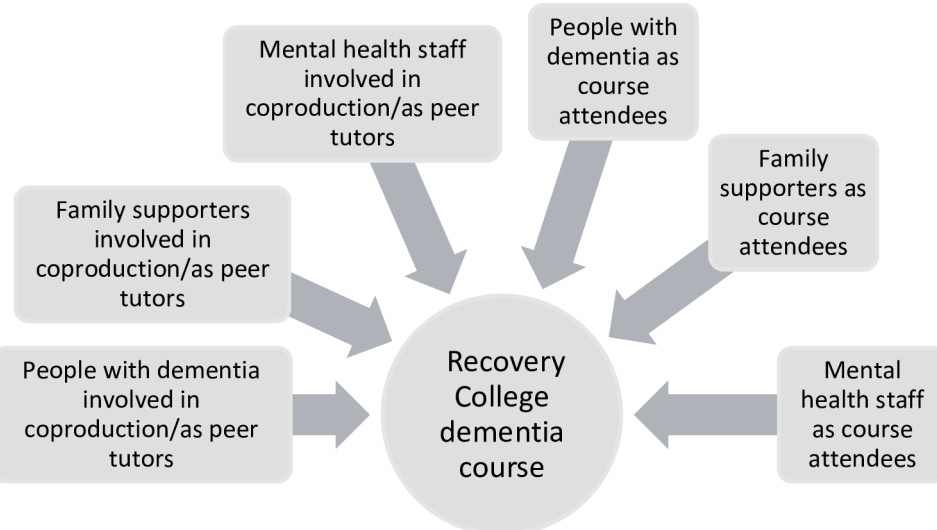

**Figure 1** Stakeholders in Recovery College courses.

socioeconomic and deprivation indices, urban/rural population mix.

► Recovery Colleges offering virtual or hybrid as well as traditional face-to-face courses.

► Courses offered by different staff groups, that is, psychologists, nurses, doctors, occupational therapists, support workers/assistant practitioners.

► Courses that involve both observable coproduction and codelivery of the course.

### Exclusion criteria

► Courses provided for or predominantly attended by family carers or staff.

► Courses not coproduced with people with dementia.

A Recovery College dementia course coproduction group typically ranges from 2 to 4 people; we plan to recruit and interview all who consent; across 5 case sites this will yield a sample of around n=15 individuals. These will include people living with dementia, family supporters and staff all involved in coproduction and codelivery of the course. Staff could be nurses, psychologists, support workers/assistant practitioners, doctors, occupational therapists, other allied healthcare professionals or service support staff (see figure 1, eg of stakeholder groups).

The usual number of attendees at Recovery College dementia courses ranges from 8 to 20 people. Across 5 case sites, this would yield a recruitment pool of between 40 and 100 people. We will purposively sample to recruit n=3–5 people per site ranging from people living with dementia, family/friend supporters and staff.

To examine service provision, we aim to recruit n=1–2 managers per site, who could be memory service managers/clinical team leaders or Recovery College leads/representative staff supporting Implementing Recovery for Organisational Change work in NHS mental health Trusts, giving a sample of 5–10 managers.

### Consent

Once confirmation of capacity and capability has been received from each participating NHS Mental Health Trust and a lead professional identified, those people coproducing the course will be consented into the study. Then as people enrol on the Recovery College dementia course, they will be advised of the planned research and invited to find out more about the study. PPI advisors have suggested the following words to use for the initial approach to each person booking a place on the course: 'Would you mind if two researchers join your course group?'. At this point for people who do not wish to be involved in the research, the lead professional will coordinate their individual booking to alternative courses/dates. We acknowledge this may cause small inconvenience to people not wanting to be involved in the observation research; however, this process was suggested by clinicians and PPI advisors and agreed by ethical review committee. See online supplemental file 2 for further details on valid informed consent processes and consent flow chart.

### Data collection

Within each case we will collect three types of data: (1) focused ethnography, (2) realist interviews and (3) documentary evidence. Data collection will take place over 18 months to provide time to capture data across one whole course.

### Focused ethnography

Focused ethnography methods use short, intensive, linked field visits.[22 23] This method is pragmatically, ethically and methodologically appropriate here in not imposing an unduly heavy burden on people. In each case study site two of the three realist evaluation researchers (LB, JW and MH) will carry out ethnographic observations, potentially with a PPI advisory group member as a coresearcher. The coresearcher may make notes if they chose but they will have a reflective conversation with the researchers after each observation event. All will be situated as 'embedded participants' among course attendees. Each case study will entail a minimum of 4-hour observation but depending on length of preparation and course, this may be up to 14 hours. The findings derived from our realist review and subsequent PPI and staff advisory group discussions will further direct the focus for observations. A postsession debrief and reflection period held with the course coproducers/facilitators will explore whether having researchers as embedded participants changed the dynamics or behaviours in the room. Researchers will transcribe their handwritten field notes up as soon as possible after the observation, within 72 hours. Materials relevant to the observation such as floor plans or course materials will be stored alongside observations.

### Realist interviews

After the course is delivered, we will conduct individual semistructured qualitative interviews with course coproducers and attendees as well as NHS managers in memory services or Recovery College staff. Interviews will be undertaken within 2 weeks of the course finishing and may be face-to-face or virtual depending on the participants preference. They will be recorded and professionally transcribed verbatim. Interview topic guides were developed based on the findings of the realist review and together with the PPI advisory group.[24] They include questions adapted to each stakeholder group; see online supplemental file 3 for topic guides. Using focused ethnography initially will provide the researchers with detailed in-depth understanding of what happens in practice, and this will then inform interviewing to interrogate areas of tension, complexity and uncertainty. If participants become distressed or anxious during the interview the researcher will pause check their ongoing agreement to the interview and continue to finish the interview. The researcher will ensure the participant is orientated not distress and if required supported by a relative or friend before leaving. A second interview will be offered to course coproducers and attendees who wish to speak specifically about meeting the needs of ethnic and cultural minority groups

**Table 1** Questions to direct analysis of data from ethnography, interviews and documentary evidence

| Relevance | Is source text section relevant to programme theory development? |
|---|---|
| Interpretation of meaning | If yes, does text content provide data that might be interpreted as functioning as context, mechanism or outcome (C, M or O)? |
| Interpretation and judgement about CMO configurations | For data that functions as C, M or O, which CMO configuration (CMOC) (partial or complete) does it belong to? Can additional data informing this particular CMOC be found either in this source or other sources? If so which sources? How does this CMOC relate to other already-developed CMOCs? |
| Interpretations and judgements about programme theory | How does this particular (full or partial) CMOC relate to the programme theory? What data from our sources inform how the CMOC relates to the programme theory? If yes, which sources provide which data? Does the construction of this specific CMOC and any supporting data, require the programme theory to be changed? |

in their courses. A brief summary of key points from the interview transcripts will be made available on request for participants to help people, particularly participants with dementia, to revisit their conversation, as was suggested by the PPI advisory group. This provides a chance for participants to change or add anything to the summary.

## Documentary evidence

We will review relevant documents such as course materials and evaluations to provide context and some triangulation with observations and interviews. For example, whether anonymised evaluations resonate with experiences discussed in interviews.

## Data analysis

All relevant data will be uploaded into NVivo,[25] a software package that supports qualitative data analysis. An iterative approach to analysis will enable emerging ideas to be refined and further explored in later case studies. We will analyse data collected using a realist logic.[20] Data coding will be deductive (informed by our initial programme theory developed from the realist review), inductive (derived from the collected data) and retroductive (making inferences about mechanisms based on interpretations of our data to infer underlying causal processes). Questions on interpretations and judgement will guide the focus of analysis as outlined in table 1.

Using a case study approach means that interpretations such as inferred mechanism of outcome can be examined and compared across cases to better explain why and how observed outcomes occur. We can compare 'successful' with 'less successful' cases of Recovery College courses, to examine how specific contexts or mechanisms have influenced outcomes observed. When using the questions in table 1, we will also use the following forms of reasoning to make sense of the data:

► Juxtaposing: for example, where data about behaviour change in one source enables insights into data about outcomes in another source.
► Reconciling: where data differ between apparently similar circumstances (ie, interviews and documentary

review), it is appropriate to seek potential explanations for why these differences have occurred.
► Adjudicating: in terms of methodological strengths or weaknesses of the data sources.
► Consolidating data: where outcomes differ in particular contexts, constructing an explanation of how and why these outcomes occur differently.

Reporting of results will be guided by Realist And MEta-narrative Evidence Syntheses: Evolving Standards.[26] The findings will be used in another work package to co design resources and materials for providers of Recovery colleges in dementia.

In summary, the aim of this realist evaluation data collection and analysis is to specify theoretical claims that are grounded in data and that may be made transferable to real world contexts. Examining interview, observation and document data alongside each other will enable us to identify and apply differing perspectives and understandings, to refine the programme theory, in relation to experiences and discussions of dementia.

## Data management

All confidential data such as consent forms and other study documentation will be archived securely for a period of 10 years in accordance with sponsor policy. Interview recordings will be transcribed and pseudonymised then destroyed. All data will have any identifying features removed. Data for analysis will be managed in NVivo.[25] To support the development of the programme theory, all study data, such as interview transcripts, observations, notes from stakeholder and researcher analysis meetings are stored in a central file and ordered by date and type. Non-identifiable data sets will be made available to other researchers at reasonable request. Data management and storage will comply with UK General Data Protection Regulations.[27]

## ETHICS AND DISSEMINATION

An important ethical concern in this study is gaining and monitoring consent from people living with dementia

who may have fluctuating capacity. We will use the principles of process consent[28]—see online supplemental file 2. Ethical approval was given by Coventry & Warwickshire Research Ethics Committee (22/WM/0215). This is a committee flagged for expertise in studies in dementia. A PPI advisory group member living with dementia attended the review meeting with the research team. The person found this an interesting experience and was active in directly addressing questions from the ethical review committee.

Learning from this research will be used in the following study work package where we will coproduce guidance and resources for groups planning or currently delivering Recovery College courses in dementia. Additional findings and learning will be disseminated through peer-reviewed publications, conferences, lay reports and via our website https://www.nsft.nhs.uk/discovery-study/.

## DISCUSSION

This ethnography draws on realist evaluation approaches to analyse, interpret and judge theoretical implications of data from naturally occurring Recovery College courses in dementia. It is likely to be the first study to report rigorously analysed findings, using a realist approach, from what happens during the coproduction and codelivery of such courses to include the perspective and actions of people living with dementia, their family and friend supporters and healthcare staff, and how they affect each other in the context of each setting. This means identifying, recording and integrating outcomes and mechanisms for the course attendees and not only for the peer tutors and staff. This evaluation of Recovery College dementia courses will provide comprehensive evidence on their place within postdiagnostic support pathways in dementia, essential for implementing evidence-based practice in Recovery College dementia courses.

**Author affiliations**
[1]School of Health Sciences, University of East Anglia, Norwich, UK
[2]School of Healthcare University of Leicester, Leicester, UK
[3]Research and Development, Norfolk and Suffolk NHS Foundation Trust, Norwich, UK
[4]Nuffield Department of Primary Care Health Sciences, Oxford University, Oxford, UK
[5]CRIPACC, University of Hertfordshire, Hatfield, UK
[6]Innovations in Dementia, Exeter, UK
[7]Faculty of Health Sciences, Department of Psychological Health and Well Being, University of Hull, Hull, UK
[8]Faculty of Health Sciences, University of Hull, Hull, UK
[9]Dementia, London, UK
[10]Research, Norfolk and Suffolk NHS Foundation Trust, Norwich, UK
[11]Older People's Services, Norfolk and Suffolk NHS Foundation Trust, Norwich, UK
[12]Nuffield Department of Primary Care Health Sciences, University of Oxford, Oxford, UK
[13]Department of Psychological Sciences, Norwich Medical School, Norwich, UK
[14]Medical School, College House University of Exeter, Exeter, UK

**Acknowledgements** We would like to thank the DiSCOVERY Partners in Research (Patient and Public Involvement) advisory group who are actively contributing at all stages of this project, guiding the researchers to always consider actions from the perspectives of people living with dementia. Special thanks to Partner in Research Martyn Gardener who worked with the researchers to defend the application at the ethics committee meeting. Thanks to members of the memory services and Recovery College staff advisory group. Thanks also to the wider DiSCOVERY study team: Charlotte Wheeler, Leanne Hague, Thomas Rhodes, Maria Sanchez and Robert Kelly. Fiona Poland and Linda Birt's time has, in part, been supported by the NIHR East of England Applied Research Collaboration.

**Contributors** JW led on writing the protocol supported by LB, FP, GW, MH, RL, CH, EM-C, EW, BT, RM, KS, CD and CF, and all contributed to protocol conception and design; LB is an expert on ethnographic methods; FP is an expert on social research methodology and community participative methods; GW is an experienced realist researcher; MH led the realist review which supported the development of this study; RL is the patient and public involvement lead supporting people with dementia as stakeholders and overseeing coproduction; BT contributed to protocol development and advised on issues relating to health inequalities; CF contributed to design and evaluation; LB and JW wrote the first draft of the article. FP, GW, MH, RL, CH, EM-C, EW, BT, CD and CF made substantial contributions to later versions.

**Funding** This study is funded by the National Institute for Health Research (NIHR) (NIHR Health and Social Care Delivery Research NIHR131676, 2022-2024). This project is affiliated with the Inclusive Involvement in Research for Practice-led Health and Social Care within the NIHR East of England, which has supported Poland and Birt's time on this study. It develops work undertaken by West in her Collaboration for Leadership in Applied Health Research & Care (CLAHRC) East of England fellowship.

**Disclaimer** The views expressed are those of the author(s) and not necessarily those of the NIHR or the Department of Health and Social Care.

**Competing interests** None declared.

**Patient and public involvement** Patients and/or the public were involved in the design, or conduct, or reporting, or dissemination plans of this research. Refer to the Methods section for further details.

**Patient consent for publication** Not applicable.

**Provenance and peer review** Not commissioned; externally peer reviewed.

**ORCID iDs**
Linda Birt http://orcid.org/0000-0002-4527-4414
Juniper West http://orcid.org/0000-0003-0802-7243
Corinna Hackmann http://orcid.org/0000-0002-4940-6998
Esme Moniz-Cook http://orcid.org/0000-0002-7232-4632
Claire Duddy http://orcid.org/0000-0002-7083-6589

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
