## [Reviewer comments · BMJ Open]

ARTICLE DETAILS

TITLE (PROVISIONAL)	Protocol for a realist evaluation of Recovery College dementia courses: Understanding co-production through ethnography
AUTHORS	Birt, Linda; West, Juniper; Poland, Fiona; Wong, Geoff; Handley, Melanie; Litherland, Rachael; Hackmann, Corinna; Moniz-Cook, Esme; Wolverson, Emma; Teague, Bonnie; Mills, Ruth; Sams, Kathryn; Duddy, Claire; Fox, Chris

VERSION 1 – REVIEW

REVIEWER	Chadborn, Neil University of Nottingham, School of Medicine
REVIEW RETURNED	22-Aug-2023

GENERAL COMMENTS	This is an important and timely study. This protocol is mainly clear and comprehensive; minor queries, mainly for clarification are detailed below. I have some broader questions which relate to assumptions for the study. I anticipate that the realist review (submitted for publication) may address these questions, but maybe the authors could elaborate or add some discussion in response to the following: There is some discussion of 'Recovery' as applied to services for people with dementia, but I would like to see some issues expanded upon, as I believe these may be important aspects to consider as contexts within CMOC. Firstly I am concerned that 'recovery' may not be concordant with people with dementia. Recovery developed from the anti-psychiatry movement; ie as an alternative 'approach' to conventional psychiatric services. For early or moderate stage dementia, many people cannot access any health services (state funded (NHS/social care) services beyond diagnosis service) and therefore 'dementia recovery' cannot really be positioned as an alternative. Instead, as the authors indicate, it is framed as complementary. However this may have led to a change in the 'approach' of the 'recovery model' as applied to dementia. This is reinforced in the language used to describe the post diagnosis situation -as the authors state: 'Support' is provided, as opposed to care. 3rd sector (charity) provision as opposed to a service (health or social care, state funded). Secondly, the authors mention "individual mental health recovery" – the specific use of the term recovery here is often misunderstood by the dementia 'community', I believe (from personal communications). People indicate that 'recovery' is a false hope, because so many messages about dementia are that it is incurable (terminal). It could be argued that the latter narrative is situated within a broader stigma, and maybe a greater emphasis on this study could be about stigma. But a secondary question
--

	could be whether 'recovery' is the optimal term to use for this population? I'd like to emphasise that in making the above points, I am prompting whether the realist evaluation could include some of these structural issues. On the other hand there is little description of which outcomes relating to individuals with dementia will be explored within the realist evaluation; briefly given as: "...outcomes for peer tutors and those attending the courses." Will the realist evaluation attempt to capture the impact the recovery college service has on individual's day-to-day living at home, or social connections etc, or will it be more focused on experience within the service itself and more direct outcomes? There is no mention of RAMESES or alternative reporting guideline for realist evaluation. Minor comments: Check date of Perkins et al 'Continuing to be me' – I think this should be 2016 p5 "Such courses align with the strategic health service aim to empower people living with dementia and their supporters to not only manage their symptoms but also to live positively or 'well' with dementia (13)." Replace with 'national dementia strategy'? Will participants be recruited from primary care? "diagnosis of dementia in English NHS mental health Trusts" – will GPs be notified (as the research relates to care plans etc held by GP)? Had the "Partners in Research" had experience of Recovery? Could people who have previously been involved in co-production of the dementia recovery colleges been recruited to join 'partners in research'? Two case study site researchers – Is two researchers sufficient to cover 5 sites? Presumably successful recruitment and data collection will be dependent on the local co-production 'team members' – can the process be described more clearly, including incentivises or remuneration for co-production team members? "Interview recordings will be transcribed and anonymised then destroyed. All data will have any identifying features removed" I would argue that it is not possible to anonymise or remove all identifying features from ethnographic data. Pseudonymised at best. Assessment of who misses out on Recovery college – ie access issues? Shock of diagnosis can lead to people 'dropping out' or turning down services at an early stage. "If at this point people do not wish to be involved in the research, the Lead Professional will coordinate their individual booking to alternative courses/dates" I'm concerned that this clause could conflict with the (usual) statement that declining to participate does not affect an individual's rights or service – because no doubt 'alternative' booking will lead to delay in access the service (ie people may feel pressure to participate or face a delay in service). Another example is in the flowchart for consent: 'Consent process completed with a personal consultee' – the flow chart seems to indicate an assumption that the consultee will advise that the
--	---

	individual will participate – ie there is no ‘excluded’ option arrow. Also, this should not be referred to as consent but consultee advice on the individual’s wishes. On a similar point: “Where people lack capacity, we will seek advice from a personal consultee (who may or may not be the person’s family/friend supporter) on what the wishes and feelings of the person might be, and whether or not they should take part.” Where a personal consultee is not a person’s supporter – on what basis would they be able to report the wishes or feelings of the person? An alternative question is how do participants give consent to the service itself (Recovery College)? Are best-interests decisions made, for example. Maybe an Independent Mental Capacity Advocate is involved? Flowchart for consent – appears to focus on individuals with dementia and does not mention care partners or staff. First attempt to obtain consent does not appear to involve an assessment of capacity? It may be mentioned in full ethics application, but what mechanisms are in place for individuals who may be triggered or upset during interviews?
--	---

REVIEWER	Palm, Rebecca University Witten Herdecke Faculty of Health, Department of Nursing Sciences
REVIEW RETURNED	31-Aug-2023

GENERAL COMMENTS	Dear authors, this is an interesting and important project and the publication of the study protocol contributes to the transparency of the realist evaluation. Therefore I strongly support the publication. The manuscript is well written and comprehensible. There are two aspects that in my opinion should be improved/ are missing:  1. Please include the RAMESES checklist for Realist Evaluation projects. 2. Please give more detailed information about your initial program theory and how it guides the evaluation process. It is important for the reader to know about your theoretical assumptions, please specify. Please make explicit, how your IPT has an influence on methods, e.g. sampling. You write: "A purposive sample will include characteristics highlighted through our realist review as most relevant for developing the programme theory". Please explain the theoretical assumptions reasoning your inclusion criteria. I would also expect to read more specified research questions that are in line with your IPT.
--

VERSION 1 – AUTHOR RESPONSE

Comments to the Author from Reviewer: 1 [Dr. Neil Chadborn, University of Nottingham]	
This is an important and timely study. This protocol is mainly clear and comprehensive; minor queries, mainly for clarification are detailed below. I have some broader questions which relate to assumptions for the study. I anticipate that the realist review (submitted for publication) may address these questions, but maybe the authors could elaborate or add some discussion in response to the following:	Thank you and we agree it is necessary to make this a standalone paper not reliant on the review paper. We have added further detail on how Rapid realist review has informed purposive sampling and include research objectives in response to reviewer 2
There is some discussion of ‘Recovery’ as applied to services for people with dementia, but I would like to see some issues expanded upon, as I believe these may be important aspects to consider as contexts within CMOc. Firstly I am concerned that ‘recovery’ may not be concordant with people with dementia. Recovery developed from the anti-psychiatry movement; ie as an alternative ‘approach’ to conventional psychiatric services. For early or moderate stage dementia, many people cannot access any health services (state funded (NHS/social care) services beyond diagnosis service) and therefore ‘dementia recovery’ cannot really be positioned as an alternative. Instead, as the authors indicate, it is framed as complementary. However this may have led to a change in the ‘approach’ of the ‘recovery model’ as applied to dementia. This is reinforced in the language used to describe the post diagnosis situation -as the authors state: ‘Support’ is provided, as opposed to care. 3rd sector (charity) provision as opposed to a service (health or social care, state funded).	We concur with your concern about the term ‘Recovery’ and this is one of the items we explore in interview (interviews topic guides added as a supplementary file 3). We add further detail explicitly exploring concerns about the term on page 5 and add reference to O Hagon (2008) and Perkins and Repper’s (2021) commentaries on the use of the word Recovery. Nonetheless the Recovery College structure is used to advertise and recruit to the dementia courses under examination so we hope the outcomes from this realist evaluation will identify any mechanisms at play in the language used.
Secondly, the authors mention “individual mental health recovery” – the specific use of the term recovery here is often misunderstood by the dementia ‘community’, I believe (from personal communications). People indicate that ‘recovery’ is a false hope, because so many messages about dementia are that it is incurable (terminal). It could be argued that the latter narrative is situated within a broader stigma, and maybe a greater emphasis on this study could be about stigma. But a secondary question could be whether ‘recovery’ is the optimal term to use for this population?	AS above we concur with your thoughts. We have added a sentence on page 4 to r make explicit the nature of recovery in this model. The ethos of the Recovery College model is not to frame the word ‘recovery’ as a clinical ‘cure’ for mental illness (10) but rather to improve outcomes for people beyond a purely medical model focus on symptom reduction, to help people rebuild meaningful, satisfying lives, despite limitations caused by mental health difficulties (10, 11).
I’d like to emphasise that in making the above points, I am prompting whether the realist	

evaluation could include some of these structural issues.	
On the other hand there is little description of which outcomes relating to individuals with dementia will be explored within the realist evaluation; briefly given as: "...outcomes for peer tutors and those attending the courses." Will the realist evaluation attempt to capture the impact the recovery college service has on individual's day-to-day living at home, or social connections etc, or will it be more focused on experience within the service itself and more direct outcomes?	The focus of the realist evaluation is on the experience of attending a 'Recovery college course' designed for people living with dementia and their family supporters. The realist evaluation explores the viewpoints of all involved across all stages from sign up delivery and perceived outcomes post course. We now include a figure to illustrate all stakeholders (page 7 Figure 1 Stakeholders in Recovery College Courses). The supplementary file 3 including topic guide more clearly provides detail on scope of questions. Specifically considering measurement of outcomes on potential efficacy of course in address social and psychological status, this is address in detail in a separate work package as part of the wider DiSCOVERY research project https://www.nsfh.nhs.uk/discovery-study as we also recognise that the majority of adult mental health Recovery Colleges collect outcome data (e.g. attendee feedback and standardised outcome measures) which varies between courses.
There is no mention of RAMESES or alternative reporting guideline for realist evaluation.	We have now added this essential reference page 10
Minor comments:	
Check date of Perkins et al 'Continuing to be me' – I think this should be 2016	Thank you, this has been corrected from 2013 to 2016.
p5 "Such courses align with the strategic health service aim to empower people living with dementia and their supporters to not only manage their symptoms but also to live positively or 'well' with dementia (13)." Replace with 'national dementia strategy'?	We have replaced text with 'national dementia strategy' as suggested.
Will participants be recruited from primary care? "diagnosis of dementia in English NHS mental health Trusts" – will GPs be notified (as the research relates to care plans etc held by GP)?	Participants are recruited through their involvement in the recovery college course in one of the 5 case sites (page No medical data is collected and GPs are not informed of a person's involvement in this realist evaluation, just as a GP is not informed if someone attends a Recovery College Course.
Had the "Partners in Research" had experience of Recovery? Could people who have previously been involved in co-production of the dementia recovery colleges been recruited to join 'partners in research'?	Within the Partners in Research group two members have experience of co-designing and running a recovery college dementia course. This information is added page 6
Two case study site researchers – Is two researchers sufficient to cover 5 sites? Presumably	There are three researchers undertaking ethnography data collection and a fourth member of

successful recruitment and data collection will be dependent on the local co-production ‘team members’ – can the process be described more clearly, including incentives or remuneration for co-production team members?	the team supporting interviews. This is added page 8. Incentives are not offered. We suggest there is sufficient detail on the planned protocol procedure but agree in the reporting of the realist evaluation we will add greater detail on the research process as ‘it happened in real life’.
“Interview recordings will be transcribed and anonymised then destroyed. All data will have any identifying features removed” I would argue that it is not possible to anonymise or remove all identifying features from ethnographic data. Pseudonymised at best.	Thank you we have changed the term used to pseudonymised page 10
Assessment of who misses out on Recovery college – ie access issues? Shock of diagnosis can lead to people ‘dropping out’ or turning down services at an early stage.	This is an important point. In part this will be addressed by questions for stakeholders on who attends and who might be missing from recovery college courses. The scope to interview ‘non attenders’ is beyond this study. The practice situation at the moment is that few NHS Trusts have courses specifically for in dementia. We have not made changes to this paper but we will keep this comment in mind when discussing limitations of the study when reporting results.
“If at this point people do not wish to be involved in the research, the Lead Professional will coordinate their individual booking to alternative courses/dates” I’m concerned that this clause could conflict with the (usual) statement that declining to participate does not affect an individual’s rights or service – because no doubt ‘alternative’ booking will lead to delay in access the service (ie people may feel pressure to participate or face a delay in service).	This was an ethical point we considered at length and discussed with people living with dementia and clinical colleagues. The conclusion was that the nature of a small group and sharing of experiences meant that ‘on course’ ethnography would necessarily involve all people in the room so this decision was made. It was agreed by a robust ethical review, which was attended by one of the Partners in Research group. We have acknowledged this tension on page 7/8
Another example is in the flowchart for consent: ‘Consent process completed with a personal consultee’ – the flow chart seems to indicate an assumption that the consultee will advise that the individual will participate – ie there is no ‘excluded’ option arrow. Also, this should not be referred to as consent but consultee advice on the individual’s wishes.	We apologise as it was not our intention to presume a consultee would be required. In many ways this goes against the ethos of the project and the research team. We have amended to flow chart and resubmit within supplementary file 2
On a similar point: “Where people lack capacity, we will seek advice from a personal consultee (who may or may not be the person’s family/friend supporter) on what the wishes and feelings of the person might be, and whether or not they should take part.” Where a personal consultee is not a person’s supporter – on what basis would they be able to report the wishes or feelings of the person? An alternative question is how do participants give consent to the service itself	We have changed the statement about to personal consultee in supplementary file 2 to ‘usually a person’s family/friend supporter’. Regarding the relationship between people with dementia and family supporters and Recovery Colleges this is beyond the scope of the DISCOVERY study and is the usual treatment offered patient accepts or rejects model.

(Recovery College)? Are best-interests decisions made, for example. Maybe an Independent Mental Capacity Advocate is involved?	
Flowchart for consent – appears to focus on individuals with dementia and does not mention care partners or staff.	We have amended the figure of the consent flow chart in supplementary file 2. Informed consent for family supports and staff runs down the left side and consent including assessment of capacity runs done the right hand side
First attempt to obtain consent does not appear to involve an assessment of capacity?	Now addressed in the flow chart
It may be mentioned in full ethics application, but what mechanisms are in place for individuals who may be triggered or upset during interviews?	We have added a sentence to data collection page 8. If participants become distressed or anxious during the interview the researcher will pause check their ongoing agreement to the interview and continue to finish the interview. The researcher will ensure the participant is orientated not distress and if required supported by a relative or friend before leaving.
Comments to the Author from Reviewer: 2 [Prof. Rebecca Palm, University Witten Herdecke Faculty of Health]	
Dear authors, this is an interesting and important project and the publication of the study protocol contributes to the transparency of the realist evaluation. Therefore I strongly support the publication. The manuscript is well written and comprehensible. There are two aspects that in my opinion should be improved/ are missing:	Thank you
1. Please include the RAMESES checklist for Realist Evaluation projects.	Thank you this important reference is now included on page 10
2. Please give more detailed information about your initial program theory and how it guides the evaluation process. It is important for the reader to know about your theoretical assumptions, please specify.	The initial programme theory has been developed through a rapid realist review which built on a Theory of Change. The review paper which will include the Theory of Change is not yet published. To help the reader understand the outcome of the review which informed the research questions and topic guides of the realist evaluation we have added objectives on page 5 which have arisen for the review. Objectives  1. To examine if the characteristics of co producers leads to difference in outcomes for attendees 2. To examine if the content and delivery of course material leads to differences in outcomes for attendees 3. To examine if the recovery college has different outcomes for different types of attendees

Please make explicit, how your IPT has an influence on methods, e.g. sampling. You write: "A purposive sample will include characteristics highlighted through our realist review as most relevant for developing the programme theory". Please explain the theoretical assumptions reasoning your inclusion criteria.	The review identified recovery college courses were co-produced by different clinical staff and not all had a co tutor living with dementia. To illustrate our theoretical assumptions, we have added objectives and specified the sampling criteria on page 7 which now supplements the inclusion criteria.
I would also expect to read more specified research questions that are in line with your IPT.	We have added research objectives page 5 based on the data emerging from the rapid realist review

VERSION 2 – REVIEW

REVIEWER	Chadborn, Neil University of Nottingham, School of Medicine
REVIEW RETURNED	30-Oct-2023

GENERAL COMMENTS	Thanks to the authors for taking care in responding fully to reviewers concerns. Amendments to the text sufficiently address all my queries.
--

REVIEWER	Palm, Rebecca University Witten Herdecke Faculty of Health, Department of Nursing Sciences
REVIEW RETURNED	03-Nov-2023

GENERAL COMMENTS	Congratulations to this precise study protocol. Revisions are well done.
--